# Pain-Free Alpha-Synuclein Detection by Low-Cost Hierarchical Nanowire Based Electrode

**DOI:** 10.3390/nano14020170

**Published:** 2024-01-12

**Authors:** Gisella M. Di Mari, Mario Scuderi, Giuseppe Lanza, Maria Grazia Salluzzo, Michele Salemi, Filippo Caraci, Elena Bruno, Vincenzina Strano, Salvo Mirabella, Antonino Scandurra

**Affiliations:** 1Department of Physics and Astronomy, University of Catania, “Ettore Majorana”, Via Santa Sofia 64, 95123 Catania, Italy; gisella.dimari@dfa.unict.it (G.M.D.M.); elena.bruno@dfa.unict.it (E.B.); salvo.mirabella@dfa.unict.it (S.M.); 2Institute for Microelectronics and Microsystems of National Research Council of Italy (CNR-IMM), Catania (University) UNIT, Via S. Sofia 64, 95123 Catania, Italy; vincenzina.strano@ct.infn.it; 3Institute for Microelectronics and Microsystems of National Research Council of Italy (CNR-IMM), VIII Strada 5, 95121 Catania, Italy; mario.scuderi@imm.cnr.it; 4Department of Surgery and Medical-Surgical Specialties, University of Catania, Via Santa Sofia 78, 95123 Catania, Italy; giuseppe.lanza1@unict.it; 5Oasi Research Institute-IRCCS, Via Conte Ruggero 73, 94018 Troina, Italy; msalluzzo@oasi.en.it (M.G.S.); msalemi@oasi.en.it (M.S.); fcaraci@unict.it (F.C.); 6Department of Drug and Health Sciences, University of Catania, Via Santa Sofia 64, 95123 Catania, Italy; 7Research Unit of the University of Catania, National Interuniversity Consortium of Materials Science and Technology (INSTM-UdR of Catania), Via Santa Sofia 64, 95125 Catania, Italy

**Keywords:** pain-free α-synuclein detection, zinc oxide nanowire, gold nanoparticles, Parkinson’s disease

## Abstract

Analytical methods for the early detection of the neurodegenerative biomarker for Parkinson’s disease (PD), α-synuclein, are time-consuming and invasive, and require skilled personnel and sophisticated and expensive equipment. Thus, a pain-free, prompt and simple α-synuclein biosensor for detection in plasma is highly demanded. In this paper, an α-synuclein electrochemical biosensor based on hierarchical polyglutamic acid/ZnO nanowires decorated by gold nanoparticles, assembled as nanostars (NSs), for the determination of α-synuclein in human plasma is proposed. ZnO NSs were prepared by chemical bath deposition (CBD) and decorated with electrodeposited Au nanoparticles (Au NPs). Then, electro-polymerized glutamic acid was grown and functionalized with anti-α-synuclein. A synergistic enhancement of electrode sensitivity was observed when Au NPs were embedded into ZnO NSs. The analytical performance of the biosensor was evaluated by cyclic voltammetry (CV) and electrochemical impedance spectroscopy (EIS), using the Fe(II)(CN)_6_^4−^/Fe(III)(CN)_6_^3−^ probe. The charge transfer resistance after α-synuclein recognition was found to be linear, with a concentration in the range of 0.5 pg·mL^−1^ to 10 pg·mL^−1^, a limit of detection of 0.08 pg·mL^−1^, and good reproducibility (5% variation) and stability (90%). The biosensor was also shown to reliably discriminate between healthy plasma and PD plasma. These results suggest that the proposed biosensor provides a rapid, quantitative and high-sensitivity result of the α-synuclein content in plasma, and represents a feasible tool capable of accelerating the early and non-invasive identification of Parkinson’s disease.

## 1. Introduction

Neurodegenerative diseases are a wide group of disorders involving neuronal cell progressive loss associated with specific clinical symptoms. Cognitive deficits, observed in conditions like Alzheimer’s disease (AD), frontotemporal dementia, dementia with Lewy bodies, and mixed (i.e., vascular plus degenerative) cognitive disorders are common features [1]. Movement disorders, such as Parkinson’s disease (PD), spinocerebellar ataxias (SCAs), and Huntington’s disease (HD) are also prevalent [2,3]. The pathophysiology of these disorders is complex, involving genetic factors in young patients, while unspecific risk factors and complex gene–environment interactions are more relevant in the aged population [4,5]. Recently, research has been focused on the molecular basis underlying neurodegenerative diseases [6], revealing aggregations of specific proteins in neuronal cells, which are also present in cerebrospinal fluid (CSF) and blood serum or plasma. For instance, beta-amyloid (βA) aggregates are associated with AD, α-synuclein with PD and other synucleinopathies, hyperphosphorylated tau proteins with AD and other tauopathies, transactive response TAR DNA-binding protein (TDP)-43 inclusions with FTD, and polyglutamine protein aggregates are associated with HD, SCAs, and other movement disorders [7]. α-synuclein plays a crucial role in the onset and course of PD and other neurodegenerative diseases [8]. Typical α-synuclein concentrations in blood plasma have been found to be 3.598 ± 2.531 and 0.157 ± 0.285 pg·mL^−1^ for PD and healthy controls (HC), respectively [9]. Of note, α-synuclein concentrations in blood plasma depend on the patient age and decline with healthy aging [10,11]. Therefore, the distinction between PD and HC requires a very sensitive analytical technique of α-synuclein.

Currently, analytical methods for the early detection of neurodegenerative biomarkers, like α-synuclein for PD patients in blood or plasma, are time-consuming and require skilled operators to handle complex and costly equipment. Therefore, a point-of-care (POC) biosensor-based method for early diagnosis, even in the preclinical stage, would be highly valuable. Commercially available determination kits of biomarkers, such as turbidimetric, nephelometric, or enzyme-linked immunosorbent assay (ELISA), have drawbacks like a high cost, being time-consuming and a low sensitivity [12]. In this scenario, nanomaterial-based electrochemical biosensors show great promise in biomedicine due to their high sensitivity, fast response and cost-effectiveness [13,14]. Disposable electrochemical biosensors have the advantage of being able to quickly monitor a patient’s clinical status, their response to therapies, and disease progression through a simple home-based self-evaluation procedure. Affordable and scalable α-synuclein detection methods need to be further developed beyond the laboratory scale and with high enough sensitivity for non-invasive discrimination between healthy and sick patients. Massey et al. proposed an organic electrolyte-gated FET aptasensor for the non-invasive monitoring of α-synuclein in saliva [15]. The authors performed the experiments using an aqueous solution of α-synuclein monomer and a supernatant solution of diluted saliva. When they used real samples of diluted saliva, some important matrix effects appear to be present at a concentration lower than 0.1 pg·mL^−1^.

Further biosensors of α-synuclein based on the electrochemical method have been proposed in the literature by Sun et al. [16], Karaboğa and co-workers [17], Tao et al. [18], Zhang et al. [19]. However, the quantification limits of the proposed biosensors were not low enough to allow the reliable use of plasma.

Recently, hierarchical nanostructures for efficient point-of-care biosensing have become increasingly interesting in the scientific community, due the possibility of developing novel biosensing technologies [20]. In this study, we propose an electrochemical biosensor of α-synuclein based on novel ZnO nano-wires arranged as a nanostar (NS). The NSs were decorated with Au NPs and functionalized with anti-α-synuclein. Nanostructures of zinc oxide, pure or in combination with other nanomaterials, have been employed in several sensing applications [21,22,23]. In particular, ZnO is an n-type semiconductor which, in the form of a nanostructure, has many advantages for biological sensing applications, such as non-toxicity, biosafety and biocompatibility. Biosensing applications exploit the strong ability of ZnO to bind biomolecules thanks to its high isoelectric point and high sensitivity due to the high surface-to-volume ratio [24,25,26]. In the proposed hierarchical nanostructure, we expect to have at least two phenomena that act synergistically to increase the sensitivity of the biosensor. One is represented by the very high surface-to-volume ratio offered by the ZnO NS. On the other hand, in the ZnO decorated with Au NPs, the noble metal has the function of generating strong electric fields, as a result of the interaction with the surrounding environment, according to surface plasmon resonance (SPR) absorption. This is due to free oscillations in the electron density. Therefore, when the semiconductor ZnO and Au NPs are embedded in the form of nanocomposite, their properties are significantly modified. In particular, a plasmon-induced electron injection has been hypothesized, thus forming a quantum sensing material [27,28]. In this way, we improved the charge transfer between the electrode and the redox probe and, therefore, the sensitivity of our biosensor. Here, we studied the synergistic enhancement of electrode sensitivity when Au NPs are embedded into ZnO NS. The novelty of this paper consists of the use, for the first time, of quantum sensing material for α-synuclein sensing applications. The active layer with a high surface-to-volume ratio, based on ZnO NSs decorated with Au NPs, effectively detects α-synuclein in clinical plasma within the optimal concentration range of 0.5 to 10 pg·mL^−1^.

We evaluated the biosensor stability and reproducibility as well as the reliable discrimination between PD and HC plasma, reaching the aim of a non-invasive but accurate clinical PD identification.

## 2. Materials and Methods

### 2.1. Materials

PET-ITO film with surface resistivity of 60 Ω/sq, L-glutamic acid (GA), potassium hexacyanoferrate (II) K_4_Fe(II)(CN)_6_, potassium hexacyanoferrate (III) K_3_Fe(III)(CN)_6_, N-Hydroxysuccinimide (NHS), *N*-(3-dimethylaminopropyl)-*N*′-ethylcarbodiimide hydrochloride (EDC), gold (III) chloride trihydrate HAuCl_4_·3H_2_O, human α-synuclein (expressed in E. coly), anti-α-synuclein (clone 1D22, Rabbit Monoclonal) and bovine plasma albumin (BSA) were supplied by Merck–Sigma Aldrich (Milan, Italy). α-synuclein and anti-α-synuclein were stored at −20 °C until used, while the respective stock solutions were prepared with phosphate-buffered saline (PBS) at pH 7.0, stored at 4 °C and used within two weeks. All the solutions were prepared by water treated in a Milli-Q system (Merck Millipore, Burlington, MA, USA), characterized by a total organic content TOC ≤ 5 p.p.b. and a resistivity of 18.2 MΩ∙cm.

### 2.2. Electrode Fabrication

Before their use, PET-ITO substrates were deeply cleaned according to the procedure reported in the literature [29]. In detail, the substrates underwent 3 phases of sonication for 10 min, in sequence, with acetone, soapy water, and Milli-Q water. Then, PET-ITO substrates were treated by an aqueous solution of H_2_O_2_, NH_3_, and H_2_O at a molar ratio of 1:1:5, at 25 °C for 90 min, in order to functionalize the surface with the -OH groups. The active sensor area of 1 cm^2^ was defined by a thin layer of Carnauba wax that was placed onto the electrode with a cotton swab. Furthermore, this step avoids the effects of the solution–air interface and isolates the unmodified portion of the ITO film. The error in the definition of the active area was quantified in less than 10%.

The aqueous dispersion of ZnO NSs was obtained by chemical bath deposition (CBD). The detailed description of ZnO NS preparation was reported elsewhere [30]. ZnO NSs were washed by Milli-Q water through decantation, dried in oven at 100 °C for 16 h to remove the adsorbed water and annealed in a furnace at 300 °C for 1 h under N_2_ flux. Then, 100 µL of aqueous dispersion, containing 2 mg·mL^−1^ of ZnO NSs, were spin-coated on the clean PET-ITO substrates at 500 rpm for 60 s and dried at room temperature. To study the effect of ZnO NSs on the electrode performance, a further batch was prepared without this material, while the other fabrication steps remained unchanged.

The biosensors were fabricated onto the PET-ITO substrate according to the main steps 1–5, depicted in Figure 1.

Au NP electrodeposition

Au NPs were electro-deposited onto ZnO NSs from 1 mM HAuCl_4_ solution in PBS, through cyclic voltammetry, by sweeping the potential from −0.2 V to −1.3 V vs. Ag/AgCl at a scan rate of 50 mV·s^−1^. The number of CV cycles were varied from 4 to 8, in order to find the best sensor response.

The electrode morphology, after Au NP decoration at various CV cycles, was characterized by FESEM (Appendix A). By increasing the number of CV cycles, an increased Au NPs coverage was obtained. Appendix A show the morphology at the sixth and tenth CV cycles of Au electrodeposition. 

Appendix A shows the first and sixth CV cycles of Au electrodeposition, at which point the process reached stabilization. Further deposition cycles did not show a substantial change in the peak assigned to the reduction of the intermediate AuCl_2_^−^ to metallic gold at about −1.1 V [31].

Appendix A reports the experimental and simulated electrochemical impedance spectra of ZnO decorated with Au NPs, obtained at various CV cycles. The inset shows the equivalent electrical circuit used for the simulations of the EIS. The Warburg element (Z_W_) allows one to analyze the mass transport resistance. The cell ohmic resistance R_Ω_ is related to the voltage drop across the solution; the C_dl_ is the double layer capacitance; R_ct_ is the charge transfer resistance that we assumed as key parameter for the definition of the biosensor response [32]. EIS characterizations show that 6 cycles of electrodeposition minimize the R_ct_ and, therefore, we used this experimental condition for the fabrication of the biosensors. To study the effect of Au NPs on the electrode performance, a further biosensor batch was prepared without this material, with the other fabrication steps remaining unchanged.

2.GA electro-polymerization

The next step was the electro-polymerization of GA, performed by 10 cycles of CV in a 0.01 M solution in PBS at pH 7.0, in the potential range from −0.8 V to 2 V. Appendix A shows the first and tenth CV cycles of GA electro-polymerization. The aim of the obtained poly-GA (PGA) is the functionalization with carboxyl groups on both Au NPs as well as ZnO NS surfaces. Beyond the tenth cycle, no change was observed in the CV and, therefore, a complete coverage of the electrode surface can be assumed. At this stage, the electrodes were stored in the dark at 4 °C.

3.Functionalization with EDC-NHS

After the electro-polymerization of glutamic acid, the electrodes underwent an incubation in aqueous solution containing 0.4 M EDC and 0.1 M NHS for 60 min at 20 °C, in order to convert carboxyl terminal groups of PGA into N-hydroxysuccinimide ester intermediate. This intermediate allows the immobilization of the anti-α-synuclein and BSA.

4.Anti-α-synuclein and BSA immobilization

After thoroughly washing with water, the electrodes were incubated first into a solution containing 100 ng·mL^−1^ of anti-α-synuclein in PBS for 60 min at 20 °C. The conditions of this step were optimized based on the literature [16]. Then, the electrodes were incubated into a 0.5 wt./vol % BSA solution in PBS to saturate the rest of unbonded sites. The electrodes were then stored in the dark at 4 °C. 

5.α-synuclein recognition and anchoring

Finally, the biosensors were employed for the determination of α-synuclein. For this purpose, the biosensors were incubated in the dark at 20 °C for 60 min with solution in PBS containing 0.5, 1, 3.5, and 10 pg·mL^−1^ of anti-α-synuclein, respectively. An optical picture of the whole electrode is shown in Appendix A.

### 2.3. Clinical Sample Preparation and Testing

A 74-year-old female patient, diagnosed according to the latest diagnostic criteria for PD [33], was recruited, along with a sex-matched HC without any neurological or neuropsychiatric disorders. The PD patient had an approximately two-year history of progressive motor symptoms and exhibited a mixed phenotype, although the rigid–akinetic component (more evident in the lower limbs) was prevalent of the tremor-related features. She did not have significant past medical conditions or relevant comorbidities, exhibited a normal cognitive functioning, and, at the time of examination, she did not take any PD-related drugs or any other psychotropic medication. The subjects were recruited at the Oasi Research Institute—IRCCS of Troina (Italy) and both the PD patient and HC provided informed consent. The study was carried out in accordance with the Declaration of Helsinki of 1964, and its later amendments.

To fulfill the specific objective, a trained nurse sampled 10 mL of peripheral venous blood from both participants through a tube without anticoagulants or other additives. As soon as the blood sample was extracted, for plasma separation each tube was maintained at room temperature, centrifuged for 10 min at 2500 rpm, then aliquoted and processed. 

### 2.4. Apparatus and Instrumentations

Morphological analyses were conducted by scanning field emission electron microscopy (FESEM) using a Gemini SUPRA 25 (Carl Zeiss GmbH, Jena, Germany). FESEM images were analyzed by ImageJ software, version 1.54g [34].

Transmission electron microscopy (TEM) analyses of ZnO NSs and Au NPs dispersed on a TEM grid were conducted using a probe aberration-corrected JEOL JEM-ARM200F microscope (JEOL USA, Inc., Peabody, MA, USA), operated at a primary beam energy of 200 keV. The microscope was operated in scanning TEM (STEM) mode, with a primary beam energy of 200 keV, and equipped with a 100 mm^2^ silicon drift detector for energy-dispersive X-ray (EDX) spectroscopy. To prepare the TEM specimens, both samples were transferred from the ITO-PET substrate onto the TEM grids using a mechanical rubbing technique.

The electrochemical characterization was carried out with a VersaSTAT 4 potentiostat (Princeton Applied research, Oak Ridge, TN, USA) at room temperature (20 °C), using a three-electrode setup, with a KCl-saturated Ag/AgCl electrode as the reference (RE) and a Pt wire as the counter-electrode (redoxme AB, Norrköping, Sweden). The step of anti-α-synuclein immobilization and the analytical response to α-synuclein were characterized using an Fe(II)(CN)_6_^4−^/Fe(III)(CN)_6_^3−^ 5 mM equimolar solution in PBS at pH 7. CVs analyses were performed in the potential range of −1.0 V to 1.5 V, at a scan rate of 10 mV·s^−1^. The intensity as well as the shift in the potential of both anodic and the cathodic peaks were measured. EIS was performed at a formal potential of 0 V, in potentiostatic mode, at 5 mV of potential amplitude from 0.1 Hz to 50 kHz. The redox probe Fe(II)(CN)_6_^4−^/Fe(III)(CN)_6_^3−^ provides information on the electrode surface modification and on its ability to sustain the electron exchange of the redox reactions, since each functionalization step and the α-synuclein recognition and anchoring increase the charge transfer resistance.

## 3. Results and Discussion

### 3.1. Structural and Electrochemical Characterization 

Figure 2a reports a plan-view FESEM image of a ZnO NS, composed of six arms highly ordered on a plane, representing the hexagonal structure of ZnO [30]. Figure 2b displays the cross-section FESEM image, revealing the high-porosity active layer formed by the self-assembly of NSs, providing a very large sensing surface. Figure 2c–e show the FESEM images at low, medium, and high magnification, respectively, of ZnO NSs decorated with Au NPs. Au NPs were randomly aggregated onto the arms of NS, with a coverage that can be varied by the number of CV cycles during the electrodeposition (Appendix A). The sample showed a good homogeneity (Figure 2c), with well-distributed decorated ZnO NSs on the electrode surface. 

Figure 2f,g present two High-Angle Annular Dark-Field (HAADF) STEM images referred to the electrode at step 1 (ZnO NSs with Au NPs) and 5 (immobilization of α-synuclein at concentration of 3.5 pg·mL^−1^), respectively. In the HAADF imaging, the observed intensities were directly proportional to the atomic number of the elements present. The application of this technique allowed us to distinguish between the ZnO wires constituting the NSs and the Au NPs. Furthermore, the identification of the ZnO NSs and Au NPs can be appreciated in EDX maps (Figure 2h). Notably, after the full electrode functionalization and α-synuclein anchoring, the EDX maps (Figure 2i) evidences the presence of a carbon layer surrounding both the ZnO NSs and Au NPs.

Figure 3a,b reports the CV and EIS, respectively, of the full electrode fabrication, from ZnO NS deposition (line 0) to α-synuclein recognition at 3.5 pg·mL^−1^ (line 5). The CV curves show the ability of the Fe(II)(CN)_6_^4−^/Fe(III)(CN)_6_^3−^ redox probe to characterize the biosensor fabrication steps. The higher the cathodic/anodic peak current, the larger the electron flux between the electrode and electrolyte. The larger the potential difference among cathodic and anodic peaks (ΔV), the higher the potential drop occurring at the electrode/electrolyte interface. An increase in the ΔV from curve 0 to curve 5 suggests again a progressive electrode functionalization. From curve 0 to curve 1, the peak current slightly increased due to the Au NP decoration of ZnO NSs leading to a higher conductivity (smaller R_ct_ in Figure 3b). Then, the CV peak currents decreased from curve 1 to 2 when PGA covered the electrode surfaces. Immobilization of anti-α-synuclein in curve 3 caused a slight increase in the anodic peak, while the intensity of the cathodic peak remained almost the same as that of curve 2. This may be due to some contribution to the oxidation of the Fe(CN)_6_^4−^ specie, rather than an electronic transfer in the redox couple Fe(II)(CN)_6_^4−^/Fe(III)(CN)_6_^3−^. Probably, the contribution came from the intermediate N-hydroxysuccinimide ester linkage species employed for the immobilization of anti-α-synuclein, which can react with Fe(II) [35]. Cyclic voltammogram 4 (anti-α-synuclein and immobilizations of BSA) shows a further decrease in peak currents due to hindered electron exchange. Worthy of notice, the cathodic and anodic peaks shift and broaden progressively from curve 1 to 5. These observables are described by the Butler—Volmer equation and are attributable to an increase in the redox overpotential and a decrease in the electronic exchange efficiency [36]. Finally, the modification of cyclic voltammogram 5 from 4 is attributable to the α-synuclein protein recognition and anchoring and can be related to its concentration. Figure 3b show the corresponding EIS recorded at the various steps. Indeed, each functionalization step increased the charge transfer resistance, as each step added a layer on top of the electrode. The change in charge transfer resistance (R_ct_) from curve 1 to 5 indicates the progressive coating onto electrode during fabrication, leading to a modification of the electron transfer ability. It should be noted that some small variation in the active area and of deposited ZnO NSs among the different electrodes was unavoidable with the present preparation technique, so that some variation of EIS spectra among tested electrodes can be observed both in terms of Rct and R_Ω_. Such variation has been found also by other authors [17].

### 3.2. α-Synuclein Determination: Sensitivity, Reproducibility and Stability of the Biosensor 

To correlate the electrochemical response to the α-synuclein concentration, we performed measurements with the biosensors before (empty circles in Appendix A) and after α-synuclein recognition (full circles in Appendix A) and anchoring. Appendix A show typical CV and EIS curves, respectively. The curve modification was ascribed to the α-synuclein recognition. Curves from 1 to 4 correspond to α-synuclein concentrations of 0.5, 1, 3.5, and 10 pg·mL^−1^ in the PBS solution, respectively.

Figure 4a reports the electrochemical impedance spectra of the biosensors after α-synuclein recognition and anchoring at different concentrations. 

Table 1 reports the charge transfer resistances of the biosensors R_ct_ and the corresponding variation ΔRct measured after α-synuclein recognition and anchoring at various concentrations. The values were obtained by simulation of the EIS spectra by the equivalent circuit model reported in the inset of Appendix A. The relative error associated with R_ct_ value was lower than 5%, taking into account the instrumental sensitivity, the fitting procedure and reproducibility (see later on).

Figure 4b shows the plot of the ΔR_ct_ as a function of the α-synuclein concentration in PBS. As the α-synuclein concentration increased, an increase in ΔR_ct_ was observed. In fact, as the α-synuclein concentration increased, more antigen–antibody immune complexes were formed, which led to a reduction in the electron transfer between the electrode and Fe(II)(CN)_6_^4−^/Fe(III)(CN)_6_^3−^ redox couple.

The data were fitted by a linear least-square regression method, according to Equation (1):(1)$$\Delta R_{ct}\left(Ω\right)=0.00931\cdot c_{\alpha -synuclein}\left(pg\cdot {mL}^{-1}\right)+0.0471$$

The calibration curve of Figure 4b clearly shows a linear response of our biosensor to α-synuclein in the concentration range of 0.5 to 10 pg·mL^−1^ (R^2^ = 0.959). Since the biosensor was designed to work with plasma, the range we investigated was appropriate to cover the α-synuclein concentration. The limit of detection (LOD = 3 S/N) was estimated to be 0.08 pg·mL^−1^. Furthermore, Figure 4b shows the data from Karaboğa et al. [17], obtained by sensors with a structure similar to that of our electrodes, but without the ZnO NSs. Their lower quantification limit started from 4 pg·mL^−1^ of α-synuclein. Furthermore, we reported the data corresponding to 10 pg·mL^−1^ of α-synuclein obtained by our electrodes prepared without Au NPs (violet dot) or without ZnO Ns (green dot), respectively. The effect of increasing the electrode response due to the simultaneous presence of Au Ns embedded in the ZnO is evident. Our experimental data demonstrate a very promising sensitivity in an interesting concentration range that can allow α-synuclein determination in plasma.

Table 2 shows a comparison of the analytical performances of our electrode with electrochemical biosensors of α-synuclein reported in the literature. Sun et al. [16] proposed a biosensor for analysis in plasma, obtaining an LOD of 14 pg·mL^−1^. Karaboğa [17] proposed a biosensor for α-synuclein fabricated onto indium tin oxide-coated polyethylene terephthalate film (PET-ITO), decorated with gold nanoparticles and electro-polymerized glutamic acid, showing a linear response to α-synuclein in the 4–2000 pg·mL^−1^ range and a limit of detection of 0.135 pg·mL^−1^. Tao et al. [18] proposed an electrochemical biosensor based on Au NPs for α-synuclein based on glassy carbon electrode modified by poly(D-glucosamine)/Au NPs/multi-walled carbon nanotubes/reduced graphene oxide, obtaining a linear response in the range of 0.05–500.00 fM and a detection limit of 0.03 fM using the square wave voltammetry (SWV) method. Zhang et al. [19] proposed an interdigitated sensor (IDE) based on SWCN and Au NPs, obtaining an LOD of 0.14 pg·mL^−1^. Some biosensors present clinical sampling (both CSF, serum or plasma); still, the testing with real human samples from Parkinson’s disease patients was missing, as well as the ability to discriminate between HC and PD samples. Our electrode showed the lowest limit of detection, a linear sensitivity in the α-synuclein concentration range of interest for Parkinson’s disease identification in plasma, and laboratory proof of discrimination between plasma coming from PD or HC tests. The obtained results of our ZnO NS-based biosensor are comparable to or even better than the literature biosensors concerning the electrochemical determination of α-synuclein, in terms of sensitivity, detection range and limit of detection. 

### 3.3. Reproducibility and Stability Test

The biosensors were tested to evaluate the reproducibility of α-synuclein determination and the stability. Figure 5a shows the results of six R_ct_ measurements after the recognition and anchoring of 3.5 pg·mL^−1^ of α-synuclein, evidencing a fairly good reproducibility with a relative error lower than 5%. For testing the biosensor stability, three disposable electrodes prepared in the same batch were tested immediately after their preparation and after storage (dark, 4 °C) for two months. In Figure 5b, we compare the ΔR_ct_ of three measurements by a fresh electrode (Electrode 1) and two electrodes (2 and 3) stored for two months. A promising retention of ΔR_ct_ of 88 and 95% for electrodes 2 and 3, respectively, was observed.

### 3.4. Clinical Samples Test

The sensitivity range of our electrodes allows the measurement of α-synuclein at concentration close to that present in human plasma samples. Nonetheless, unwanted interaction with human proteins could hamper the effective α-synuclein detection. Thus, we used two different plasma samples extracted from the human blood of a PD patient and of a HC one. 

Figure 6a,b show the impedance spectra before and after the recognition step with the plasma of HC and PD patient. Furthermore, the addition method (Figure 6c) was used to confirm the results of the PD patient and to assess the presence of any matrix effects expected, since the complexity of the plasma composition and the presence of several proteins. For this purpose, α-synuclein was added to HC plasma sample at a concentration of 1 pg·mL^–1^.

The variation in charge transfer resistance after α-synuclein immobilization was significant, thus also confirming the responsiveness of the electrode in a real plasma sample. We reported the results of clinical tests in Figure 7 where the calibration curve (red line) of Figure 4b was reported together with the measured ΔR_ct_ in PD plasma, HC plasma, and HC plasma added with α-synuclein up to a final concentration of 1 pg·mL^−1^. These data were reported on the calibration line in correspondence with the ΔR_ct_ values (horizontal arrows) measured by our biosensors. By using the calibration curve, we converted the ΔR_ct_ values into an α-synuclein concentration. The HC sample test induced a ΔR_ct_ of 20 Ω, which corresponds to an α-synuclein concentration less than 0.2 pg·mL^−1^, in good agreement with the values of the healthy patient. The PD patient test exhibited a ΔR_ct_ of 100 Ω, corresponding to an α-synuclein concentration of about 0.9 pg·mL^−1^, as expected for patients affected by the Parkinson’s disease. It is important to underline that our biosensor clearly discriminated between plasma coming from the HC or PD patient. The difference in ΔR_ct_ was well larger than the experimental error, leading to a reliable discrimination. Indeed, to assess the matrix effect, we added 1 pg·mL^−1^ of α-synuclein into the HC sample, and found a ΔR_ct_ of 120 Ω, corresponding to an α-synuclein concentration of about 1.1 pg·mL^−1^, as expected. The undiluted plasma did not interfere with the quantitative assay, thus making the electrode suitable for α-synuclein determination in real plasma. We can conclude that the potential interferences coming from plasma proteins are quite low and do not affect the biosensor response. 

## 4. Conclusions

We successfully developed the first highly sensitive α-synuclein electrochemical biosensor capable of discriminating between healthy and PD patients. The biosensor was based on a hierarchical nanostructure consisting of ZnO NSs decorated with Au NPs and functionalized with anti-α-synuclein. The sensing performances were evaluated through the charge transfer resistance, measured in the impedance spectroscopy, using a Fe(II)(CN)_6_^4−^/Fe(III)(CN)_6_^3−^ probe. We showed the synergistic enhancement of electrode sensitivity when Au NPs were embedded into ZnO NSs. From the impedance spectra, the variation in the charge transfer resistance of the electrode before and after α-synuclein immobilization was shown to vary linearly with the α-synuclein concentration in the range of 0.5 pg·mL^−1^ to 10 pg·mL^−1^. The limit of detection was estimated to be 0.08 pg·mL^−1^. The biosensor reproducibility was very good (5% variation) and a fair retention (about 95%) of charge transfer resistance was demonstrated after two months of storage. The ZnO NS-based biosensor was also tested with clinical samples coming from HC and PD patient plasma, being shown to easily discriminate between them. The matrix effects related to plasma were found to be marginal. Our ZnO NS-based biosensors represent a non-invasive, rapid and reproducible diagnostic and prognostic tool towards Parkinson’s disease identification. Translationally, because of its very low LOD, it could be feasibly advanced towards a POC device.

## Figures and Tables

**Figure 1 nanomaterials-14-00170-f001:**
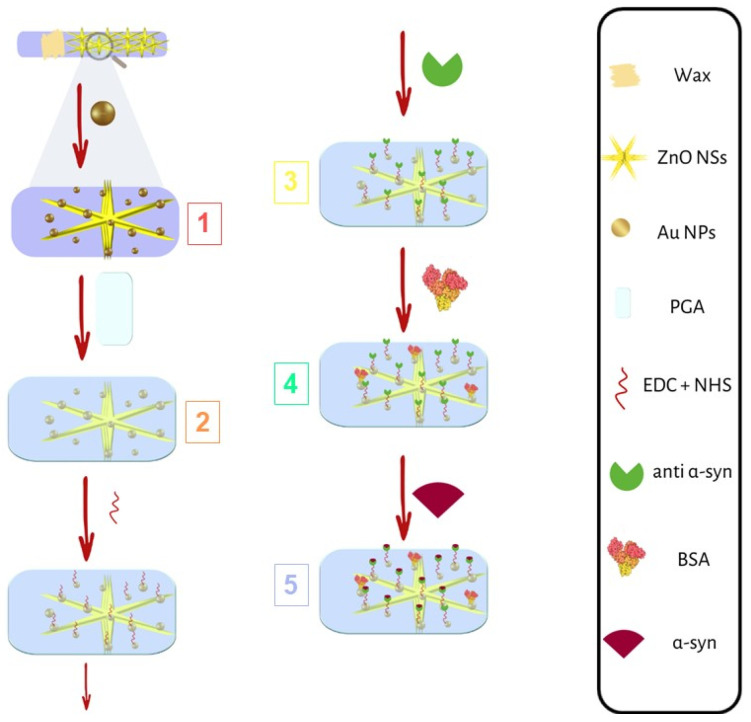
Main steps of biosensor fabrication: (1) Au NP electrodeposition; (2) GA electro-polymerization; (3) anti-α-synuclein immobilization; (4) BSA immobilization; and (5) α-synuclein recognition and anchoring.

**Figure 2 nanomaterials-14-00170-f002:**
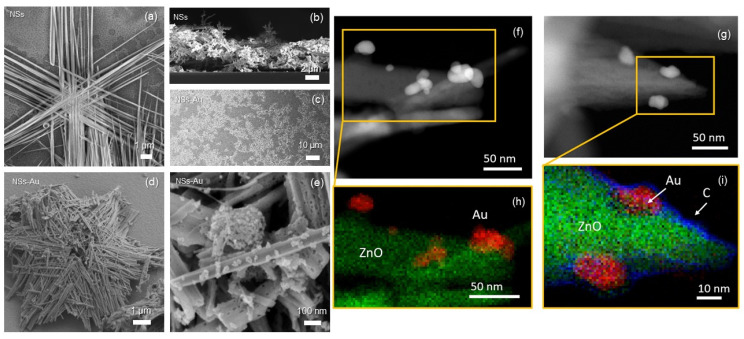
Field emission scanning electron microscopy images of ZnO nanowires assembled as a nanostar (NS) in (**a**) plan and (**b**) cross-sectional views; NSs decorated with Au NPs at (**c**) low- (**d**) medium- (tilted view), and (**e**) high-magnification plan views. NSs were spin-coated on a Si substrate for imaging. NSs are made of 5 µm long arms, each one composed of entangled parallel wires, showing a remarkable surface-to-volume ratio. HAADF STEM images of ZnO NSs decorated with Au NPs (**f**) before and (**g**) after α-synuclein immobilization; (**h**,**i**) corresponding EDX elemental maps.

**Figure 3 nanomaterials-14-00170-f003:**
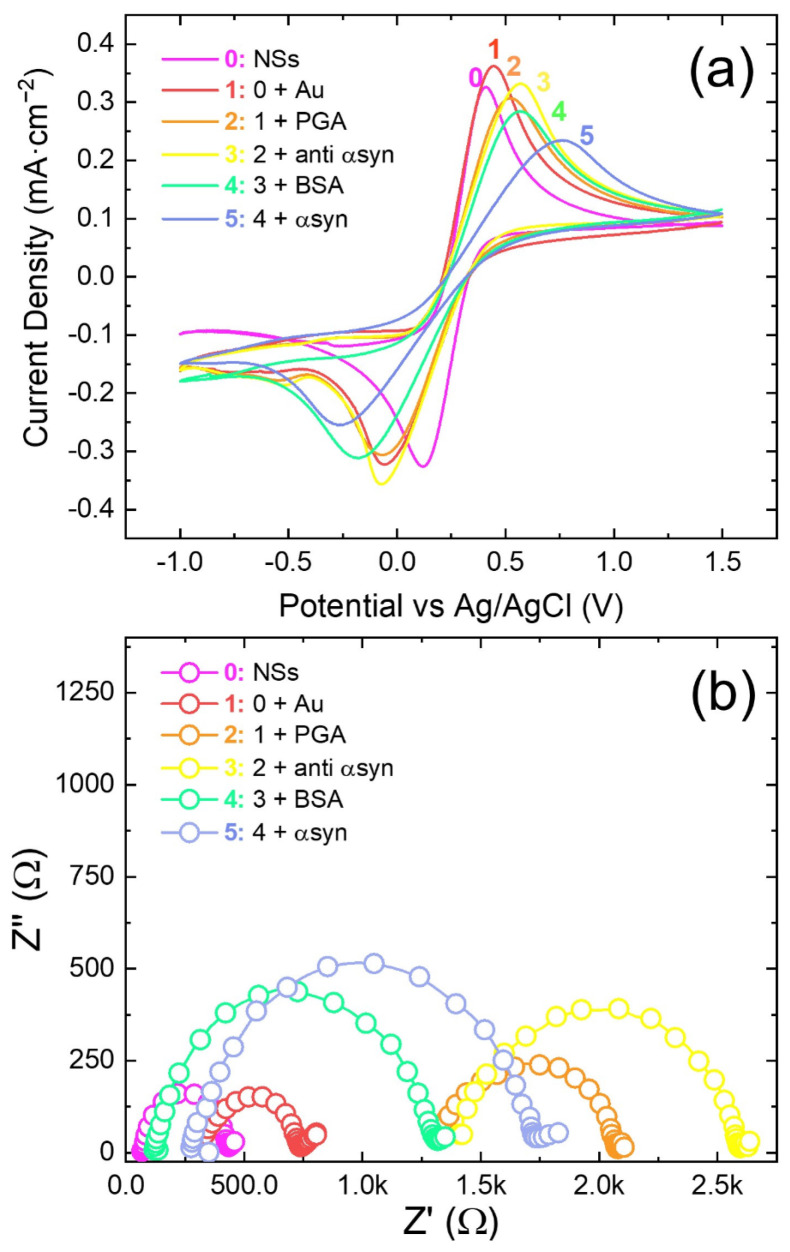
(**a**) Cyclic voltammograms and (**b**) corresponding electrochemical impedance spectra of the biosensor at the various fabrication steps: (0) NSs; (1) ZnO NSs + Au NPs; (2) NSs + Au + PGA; (3) ZnO NSs + Au NPs + PGA + anti-α-synuclein; (4) NSs + Au + PGA + anti-α-synuclein + BSA; and (5) ZnO NSs + Au NPs + PGA + anti-α-synuclein + BSA + α-synuclein. The CVs were recorded at 10 mVs^−1^; the EIS curves were recorded at a formal potential of 0 V; the α-synuclein concentration was of 3.5 pg·mL^−1^. Conditions: redox probe Fe(II)(CN)_6_^4−^/Fe(III)(CN)_6_^3−^ 5 mM equimolar solution in PBS at pH 7.

**Figure 4 nanomaterials-14-00170-f004:**
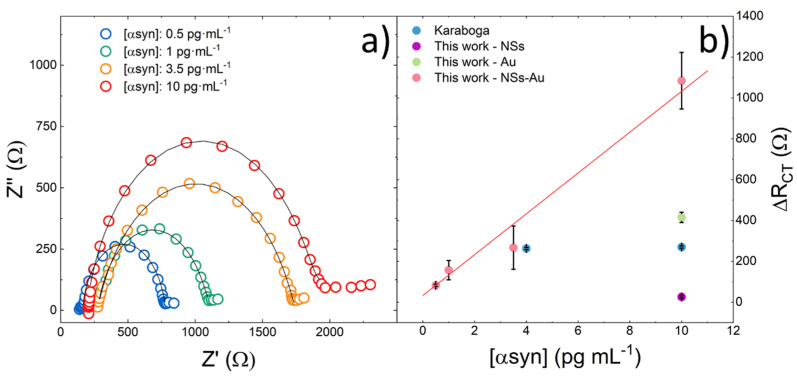
(**a**) Experimental (dots) and simulated (black lines) impedance spectra (EIS) in the frequency range of 0.2 to 5·10^4^ Hz of the electrodes after α-synuclein recognition and anchoring at different concentrations. The simulation was performed by the equivalent circuit reported in the inset. The EIS were recorded at a formal potential of 0 V. A clear increase in the R_ct_ was observed when increasing the α-synuclein concentration; (**b**) calibration curve showing linear behavior of ΔR_ct_ in the explored range of the concentration of α-synuclein in the PBS solution. The blue dots refer to data calculated from Karaboğa [17]; the violet dot refers to the sensors prepared without Au NPs; and the green dot refers to the sensors prepared without ZnO Ns.

**Figure 5 nanomaterials-14-00170-f005:**
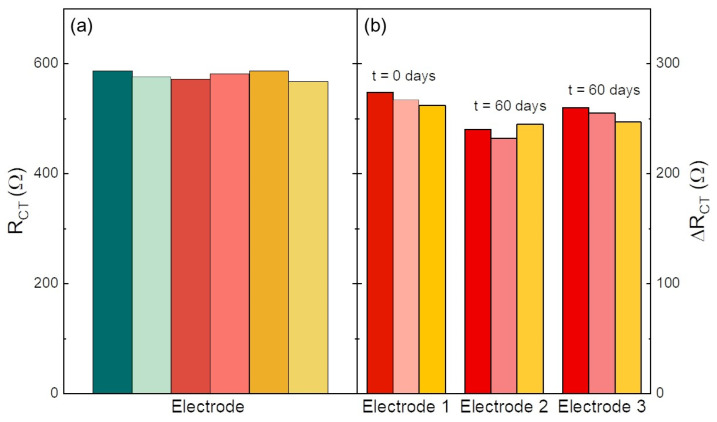
(**a**) Reproducibility test performed at a concentration of 3.5 pg·mL^−1^ of α-synuclein. The various colors refer to measurements on six electrodes; (**b**) electrode response after 60 days of storage in dark at 4 °C. α-synuclein concentration of 3.5 pg·mL^−1^. ΔR_ct_ retention after storage was of 88 and 95%. The various colors refer to measurements on three electrodes (**b**).

**Figure 6 nanomaterials-14-00170-f006:**
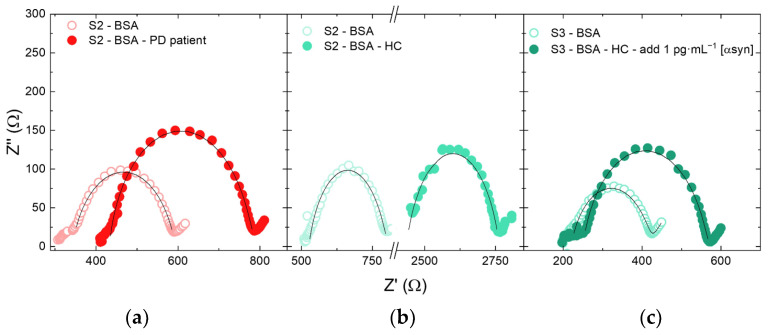
Experimental (dots) and simulated (black lines) impedance spectra (EIS) in the frequency range of 0.2 to 5·10^4^ Hz of the biosensors before and after incubation with (**a**) PD plasma and (**b**) HC plasma; (**c**) before and after incubation with control plasma added with 1 pg·mL^−1^ of α-synuclein. The corresponding ΔR_ct_ values are reported in Figure 7.

**Figure 7 nanomaterials-14-00170-f007:**
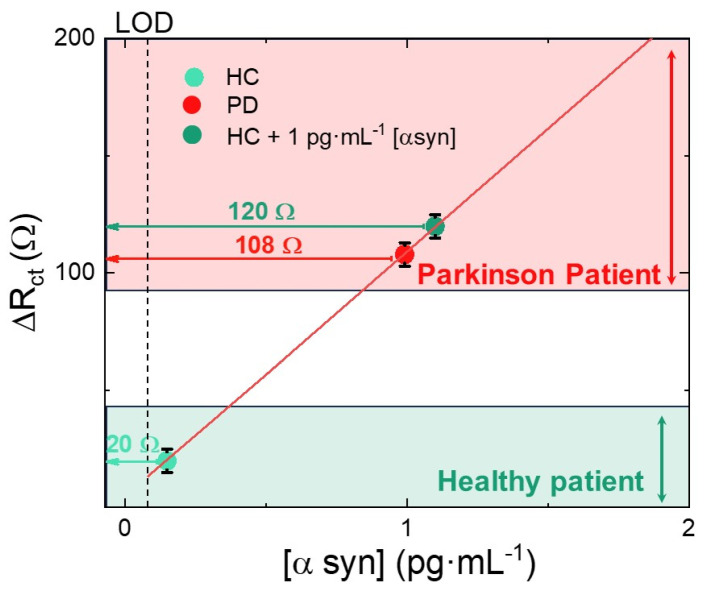
R_ct_ variation of the biosensor after clinical test in plasma coming from patient with Parkinson’s disease (PD), healthy control (HC), and HC added with 1 pg·mL^−1^ α-synuclein. Clinical samples data were placed onto the calibration line of the biosensor in correspondence of their ΔR_ct_ (ordinate axis), in order to calculate the corresponding α-synuclein concentration. Red and green boxes display the ΔR_ct_ ranges of PD and HC, respectively.

**Table 1 nanomaterials-14-00170-t001:** Parameters of the EIS simulation by the equivalent circuit model reported in Figure 5a, before (BSA step) and after α-synuclein recognition and anchoring at concentrations from 0.5 to 10 pg·mL^−1^.

α-Synuclein Concentration pg·mL^−1^	R_Ω_ Ω	R_ct_ Ω	ΔR_ct_ Ω	C_dl_ µF	Z_w_ Ω·s^−0.5^
(BSA step)	411.7	546	-	10.1	27.2
0.5	163	628.9	82.9	10.5	25.5
(BSA step)	170	655	-	9	14.5
1	276	812	157	11	44
(BSA step)	110	1194	-	3.8	47
3.5	273	1461	267	3.4	19
(BSA step)	399	605	-	12.3	21
10	192	1691	1086	3.2	25

**Table 2 nanomaterials-14-00170-t002:** Comparison of some analytical performances of α-synuclein biosensors in the literature.

Active Material	Detection Method	Sample	Detection Range pg·mL^−1^	Limit of Detection pg·mL^−1^	Testing with Real Human Samples from PD and HC Patients	Reference
Organic Electrolyte-Gated FET Aptasensor	Electrolyte-gated field-effect transistor	DI water solution of α-synuclein; diluted saliva	10^−4^–10^4^	10^−5^	No	[15]
Au NPs	EIS	Plasma (addition method)	7·10^3^–70·10^3^	14	No	[16]
Au NPs	EIS	CSF	4–2000	0.135	No	[17]
MWCNTs/Au NPs	SWV	HC Plasma (addition method)	0.7–700	0.4	No	[18]
Au-nanourchin coniugated SWCN	IDE sensor, voltammetry	Solution of α-synuclein	0.14–14	0.14	No	[19]
ZnO NSs + Au NPs	EIS	PD and HC plasma, direct and addition method	0.5–10	0.08	Yes	This work

## Data Availability

Data are available on request to authors.

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
