# Peer review of "Pain-Free Alpha-Synuclein Detection by Low-Cost Hierarchical Nanowire Based Electrode"

_nanomaterials, 2024, doi:10.3390/nano14020170_

Round 1

Reviewer 1 Report

Comments and Suggestions for Authors

The manuscript, Pain-free alpha-synuclein detection by low-cost hierarchical nanowires based electrode, by Scandurra and co-workers demonstrates interesting work in the direction of biosensing. The following are few remarks from this referee: 

-Abstract reflects the findings reported int he manuscript.

The introduction is also appropriate as the importance and literature background have been appropriately highlighted. Just the novelty part at the end should be clearly rewritten.

The experimental section is detailed and convincing too.

-What is the growth mechanism behind the 6-armed morphology of ZnO nanostructures? It is controlled?

-The digital photograph of the experimental C-V cycle should be shown in Figure 3. What is the reason behind the reduced peak height in the C-V plot in Figure 3a (1 to 5). Figures 3 and 4 can be combined together in one panel.

-The other results and discussions are appropriate. Authors are suggested to include a tentative sensing mechanism.

The conclusion is adequate.

-The manuscript needs to be checked for English and also further relevant citations can be included, for example, Biosensors and Bioelectronics: X, 2022, 12, 100284;  Materials Today Electronics 6, 2023, 100067; Materials Today 50, 2022, 533-569; and many others. 

Comments on the Quality of English Language

Minor

Author Response

To: Editor of Nanomaterials

Revision of the Manuscript Number: 2809127: Pain-free alpha-synuclein detection by low cost hierarchical nanowires based electrode

We would like to thank the Editor for the chance to improve our manuscript. We also acknowledge the unbiased and solid review of our manuscript done by the two Reviewers. We sincerely wish to thank them for the comments which stimulated us in developing it.

In fact, based on the specific comments and on the weak points identified, we deeply revised our manuscript in several parts which are now stronger and more solid. We thus believe it is now acceptable for publication in Nanomaterials.

In the following pages the Reviewer’s report is reported in italic, interrupted by indented insertions with our responses (Author reply) and relative changes (in red), if any, done throughout the manuscript at the point of the page number indicated. We hope this response format is clear enough.

 Reviewer #1

The manuscript, Pain-free alpha-synuclein detection by low-cost hierarchical nanowires based electrode, by Scandurra and co-workers demonstrates interesting work in the direction of biosensing. The following are few remarks from this referee:

-Abstract reflects the findings reported in the manuscript.

The introduction is also appropriate as the importance and literature background have been appropriately highlighted.

Just the novelty part at the end should be clearly rewritten.

Author reply: to cope with this reviewer comment, we add a new sentence through the lines 106-110 that explain better the novelty of our paper.

The experimental section is detailed and convincing too.

-What is the growth mechanism behind the 6-armed morphology of ZnO nanostructures? It is controlled?

Author reply: the growth mechanism of the 6-armed ZnO nanostructures will be reported in a future paper by the authors. Meanwhile, some details on the synthesis were reported by the authors in previous works [Strano V. and co-workers, Chemosensor, 2019] and Di Mari G.M. and co-workers, Nanomaterials, 2022, 12, 1–13 https://doi.org/10.3390/nano12152588 that has been cited in this paper as reference [30].

Briefly, in the first step, zinc nitrate is added to hexamethylenetetramine, and the latter coordinates the Zn2+ ions, thereby controlling the concentration of free zinc ions in solution. Once the fluoride is added in the form of ammonium fluoride, it, attracted by the positive charge of zinc, binds with the cation. The steric hindrance of the neutral HMTA ligands, nevertheless, hampers the growth of the nanosheets, but nonetheless leads to the formation of a distinct structure from that of a typical rod obtained through the reaction of Zn(NO3)2 and HMTA. The structure generated consists of multiple strips, orderly arranged on the plane forming a nanostar.

-The digital photograph of the experimental C-V cycle should be shown in Figure 3. What is the reason behind the reduced peak height in the C-V plot in Figure 3a (1 to 5). Figures 3 and 4 can be combined together in one panel.

Author reply: thank you to the reviewer for this question. We believe that the addition of a digital photograph of the experimental C-V in Fig. 3 is redundant, since the C-V are already shown both in the paper and in the SI in Figures 3a and S5. However, to cope with this reviewer’s request we attached in this reply the screenshot of the typical acquired CV.

 The reasons why the peak heights of the CV are reduced ongoing from CV 1 (Au NPS growth) to CV 5 (α-synuclein immobilization) is explained in the text through the lines 281 to 295 of the manuscript. Each functionalization step adds an insulating layer that increases the thickness of the isolation material between the electrode and the solution. Thus, current exchange of the redox couple is reduced and the separation between anodic and cathodic peaks increases due to the overpotential contribution. This behavior is described by the Butler-Volmer equation. Moreover, we prefer to leave the Figure 3 and 4 as in the actual form.

 -The other results and discussions are appropriate. Authors are suggested to include a tentative sensing mechanism.

Author reply: the sensing mechanism of our electrode is explained by the data of Figures 3a, 3b and 4b and through the lines 227-304 and 339-379.  The recognition and anchoring of the α-synuclein by its antibody, produce the modification of cyclic voltammogram and the charge transfer resistance between electrode and solution, measured through the redox probe Fe(II)(CN)64−/Fe(III)(CN) 63−. The variation of the charge transfer resistance is proportional to the α-synuclein concentration.

The conclusion is adequate.

-The manuscript needs to be checked for English and also further relevant citations can be included, for example, Biosensors and Bioelectronics: X, 2022, 12, 100284; Materials Today Electronics 6, 2023, 100067; Materials Today 50, 2022, 533-569; and many others.

Author reply: thank you for this comment. We checked for English that now has been improved. Moreover, we add the three good references suggested by this reviewer. We renumbered all the reference list.

Reviewer 2 Report

Comments and Suggestions for Authors

This work presents a clear research about the nanomaterial relative electrode for electrochemical measurements on the PD's biomarker sensing. The writing is clear and idea is acceptable. Some comments are listed below for authors to improve quality of whole manuscript. Current decision of reviewer is "minor revision".

1. Picture of whole electrode is suggested to provide. How to define a certain sensing area? manually or by machine?

2. How to confirm the uniform of nanostructures on electrode surface?

3. How about the selectivity to other biomarkers?

4. Due to the limitation of impedance, background solution or some interference ions can play a dominated factor. How did authors fix this concern?

Author Response

To: Editor of Nanomaterials

Revision of the Manuscript Number: 2809127: Pain-free alpha-synuclein detection by low cost hierarchical nanowires based electrode

We would like to thank the Editor for the chance to improve our manuscript. We also acknowledge the unbiased and solid review of our manuscript done by the two Reviewers. We sincerely wish to thank them for the comments which stimulated us in developing it.

In fact, based on the specific comments and on the weak points identified, we deeply revised our manuscript in several parts which are now stronger and more solid. We thus believe it is now acceptable for publication in Nanomaterials.

In the following pages the Reviewer’s report is reported in italic, interrupted by indented insertions with our responses (Author reply) and relative changes (in red), if any, done throughout the manuscript at the point of the page number indicated. We hope this response format is clear enough.

 Reviewer #2

 This work presents a clear research about the nanomaterial relative electrode for electrochemical measurements on the PD's biomarker sensing. The writing is clear and idea is acceptable. Some comments are listed below for authors to improve quality of whole manuscript. Current decision of reviewer is "minor revision".

  1. Picture of whole electrode is suggested to provide. How to define a certain sensing area? manually or by machine?

Author reply: the sensing area was defined manually by applying a wax. To cope with this reviewer’s comment, a photo of the whole electrode has been included in the SI as new Figure S4.

  1. How to confirm the uniform of nanostructures on electrode surface?

Author reply: the uniformity of active layer was guarantee by spin coating process of the ZnO NSs deposition onto PET-ITO. The uniformity of active layer was checked by field emission scanning electron microscopy (FESEM). Figure 2c shows a low magnification FESEM picture of the ZnO NSs distribution onto the PET-ITO substrate.

  1. How about the selectivity to other biomarkers?

Author reply: at the actual stage of biosensor development, we did not perform selectivity test with others biomarker. However, basing on the literature results the biosensor has a good selectivity since it is based on the recognition of α-synuclein protein by specific antibody anchored to the ZnO NSs AuNPs hierarchical structures. Furthermore, it has been tested with real human plasma that contains numerous proteins and we did not observe any matrix effect.

 Due to the limitation of impedance, background solution or some interference ions can play a dominated factor. How did authors fix this concern?

Author reply: we found that our electrode is poorly sensitive to the matrix effects even when we used a real human plasma. It is known that the effects due to the background solution or to some interference ions mentioned by the reviewer, in agreement with the literature, mainly influence the RΩ i.e. the resistance attributed partly to the solution and partly to the electrode body, while for our quantification purposes we used the change in charge transfer resistance ΔRct, which depends on the layer that incorporates the α-synuclein.
